# Stress and Resilient Coping among Nurses: Lessons Learned from the COVID-19 Pandemic

Hussein M. Magdi [ID]

The Department of Psychiatric/Mental Health Nursing, Faculty of Nursing, Beni-Suef University, Beni-Suef 62521, Egypt; magdi.hussein@nursing.bsu.edu.eg

**Abstract:** (1) Background: the COVID-19 pandemic is still taking over the world, and with new virus strains, the burden on the health systems and healthcare teams is yet to end. The study explored stress and employed resilient coping measures among nurses working with COVID-19 patients. (2) Methods: the current study used a convergent triangulation design using a qualitative and quantitative approach, with a sample of 113 nurses working at Beni-Suef University Hospital between November 2021 and April 2022 using an online form. Data were collected using a sociodemographic characteristics questionnaire, the Nursing Stress Scale, Brief Resilient Coping Scale, and a qualitative interview. (3) Results: the studied nurses experienced severe stress, with moderate-to-low resilient coping in total. There was a strong negative highly statistically significant correlation between stress and coping. Qualitative data analysis showed that attention to the spiritual side, communicating with friends and family, accepting the current situation, and helping others in need were key themes that emerged to moderate studied nurses' stress. (4) Conclusions: nurses who work with COVID-19 patients experience high levels of stress regardless of their characteristics; spiritual coping strategies, communication with family, and helping others were the dominant resilient coping approaches employed by nurses to moderate stress.

**Keywords:** stress; resilient coping; nurses; COVID-19; pandemic

## 1. Introduction

The COVID-19 pandemic started in December 2019 in China and then later spread all over the world, which forced governments everywhere to employ isolation measures and social distancing approaches [1,2]. The Egyptian government implemented a lockdown, social distancing, mandatory preventive measures, and curfews nationwide by mid-March 2020 as a rather early response to contain the spread of the COVID-19 infection like most of the world's countries [3]. Regardless of the remarkable efforts to contain the preliminary transmission of the virus, a notable acceleration in confirmed COVID-19 cases was observed in the daily reports issued by the Ministry of Health and Population during the early months of the pandemic. It was not until the first of August that Egypt recorded below 200 new COVID-19 cases for the first time since the beginning of the pandemic [4].

During the whole pandemic time, healthcare facilities designated for quarantine and treatment of COVID-19 cases were not properly equipped for such purposes; mental and physical strain were reported by most healthcare staff working in such facilities, including excessive fear and anxiety about the health of self, family, and friends, changes in sleep and eating patterns, and worsening of existing health problems. Nonetheless, there was a sense of responsibility [5], which in most cases lead to mental exhaustion [6]. In addition, there was a rise in COVID-19 infection cases and deaths among healthcare team members around the country. With a high prevalence of environmental stressors already in place and the coexisting pandemic, it is quite common for nurses to suffer work-related stress [7].

Stress is a common term indicating activities thought to promote a range of physical and mental conditions [8]. It is any internal or external cue in the physical, social, or

psychological environment which affects the balance of an individual [9]. Coping is defined as a fundamentally cognitive and consequently behavioral strength employed to control internally and/or externally originated challenges that are considered to exceed an individual's resources [10].

Staff coping with such challenges use various unknown strategies, which are mostly individualized and motivated by individual experiences, education levels, and socially available resources [11]. As described in previous research, minimizing the negative effects of a pandemic can be established with the investigation of coping strategies and factors promoting self-resilience [12,13]. Therefore, the current research employed a mixed approach to achieve the aim. The current study aimed to first evaluate stress and then explore coping strategies experienced by nurses during the COVID-19 pandemic, considering their statements.

## 2. Materials and Methods

### 2.1. Design

A convergent triangulation research design (Figure 1) was employed to achieve the current research aim [14]. In triangulation, multiple methodological strategies, datasets, and research theories are used to strengthen the validity and credibility of research findings [15].

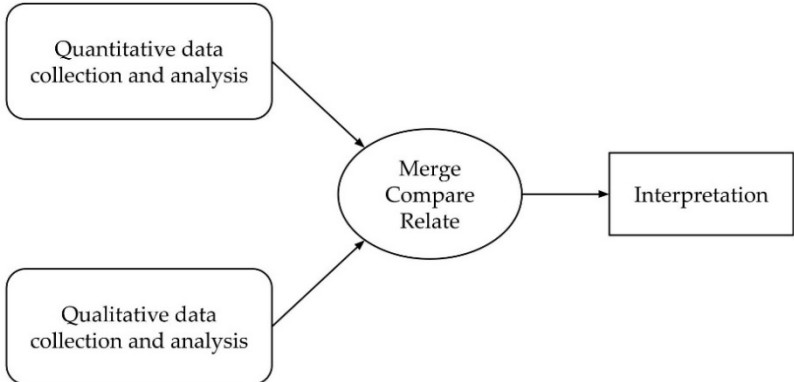

**Figure 1.** Convergent Triangulation Research Design: Convergence Model [16].

### 2.2. Sample and Setting

Data collection took place at Beni-Suef University Hospital in Beni-Suef, Egypt, between November 2021 and April 2022. The university hospital in Beni-Suef was established in 1995, and is affiliated with the Faculty of Medicine, Cairo University branch in Beni-Suef currently affiliated with Beni-Suef University (since 2005) containing 472 beds serving alongside other governmental hospitals and medical centers for the people of Beni-Suef Governorate with around twenty-five specialized clinics and departments.

The study sample was purposive and consisted of 113 nurses. Study participants were asked to consent to the use of the data collected, were informed about the anonymity and confidentiality of the collected data, and told that it would only be used for research purposes. After giving their consent, participants filled out the questionnaire online using Google Forms™ due to a lack of appropriate meeting setting at the hospital and lack of time. The qualitative interview part was conducted over descriptive, open-ended questions, phone calls, or via social media platforms. Exclusion criteria included having a diagnosed mental or physical illness.

### 2.3. Data Collection Tools

The online questionnaire was established based on the literature review of related research. A pilot study was carried out to examine and adjust data collection tools and establish a guide for qualitative questions. Quantitative data collection tools included the three questionnaires as follows:

### 2.3.1. Quantitative Data Collection Tools

- Sociodemographic characteristics questionnaire: It was developed by the researcher after reviewing the related literature. It includes items such as age, gender, education, and the number of household members, and information about previous COVID-19 infection.
- Nursing Stress Scale (NSS) [17]: It consists of 34 items that describe situations that have been identified as causing stress for nurses in the performance of their duties. The scale consists of seven subscales: death and dying, conflicts with physicians, inadequate preparation, lack of support, conflicts with other nurses, workload, and uncertainty concerning treatment. Each item's response ranges from never (score 0) to very frequently (score 4) with higher scores indicating more stress.
- Brief Resilient Coping Scale (BRCS) [18]: It consists of four items designed to identify propensities for highly adaptive stress resilient coping. Each item's response ranges from strong disagreement (score 0) to strong agreement (score 4) with higher scores indicating a tendency to reframe the influence of stressors by affirming the power of positive ways to balance possible failures.

### 2.3.2. Qualitative Data Collection Tools

Qualitative data questions guide: as qualitative description methodology dictates, the interview questions guide was used as a starting point to learn about participants' experiences and events during the COVID-19 pandemic.

* Do you feel any positive or negative emotions at work? Explain?
* How did you cope with work-related stress? Explain?
* What resources supported you during periods of having negative feelings? Explain?

### *2.4. Statistical Analysis*

The collected data were organized, tabulated, and statistically analyzed using SPSS software (Statistical Package for the Social Sciences, version 25) (Chicago, IL, USA). A Kolmogorov–Smirnov goodness-of-fit test was performed to indicate if the data collected were normally distributed and turned out to be not statistically significant for the Nursing Stress Scale (KS 0.075, P 0.163) and highly statistically significant for the Brief Resilient Coping Scale (KS 0.100, P 0.007). Comparison between variables was evaluated using Spearman's Correlation Coefficient. Degrees of significance (Sig.) of results were considered significant at $p$-value $\leq 0.05$ and highly significant at $p$-value $\leq 0.01$. Internal consistency was measured using Cronbach's Alpha and was considered high for both the Nursing Stress Scale and Brief Resilience Coping Scale (0.933, 0.912, respectively). Qualitative data were transcribed and analyzed using social studies thematic inductive and deductive evaluation to search and isolate patterns [19].

### 3. Results

### *3.1. Quantitative Data*

The current study enrolled 113 participants. As shown in Table 1, of these, 50.4% were female, 34.5% were between 25 and 35 years old, followed by 29.2% who were less than 25 years of age, and 54.0% had a university education (bachelor's in nursing). The number of household residents ranged between four and six members in 49.6% of the studied sample, and 54.9% of the studied sample highly abided by social distancing and safety precautions regarding COVID-19. Regarding COVID-19 infection, 81.4% of the studied sample had no documented COVID-19 infection among their friends, family, or coworkers. Additionally, sociodemographic characteristics collected were non-statistically significant with regard to the characteristic's subgroups and Nursing Stress Scale and brief resilience coping scale ($p$-value > 0.05).

**Table 1.** Sociodemographic Characteristics' Frequencies and Relation with Nursing Stress Scale (NSS) and Brief Resilience Coping Scale (BRCS) among Studied Sample. (*n* = 113).

| Sociodemographic Characteristic | | No. (%) | NSS | | BRCS | |
|---|---|---|---|---|---|---|
| | | | Mean ± SD | Test [a] (Sig.) | Mean ± SD | Test [a] (Sig.) |
| Gender | Male | 56 (49.6) | 66.93 ± 22.02 | 0.986 (0.326) [b] | 6.54 ± 4.13 | 1.976 (0.051) [b] |
| | Female | 57 (50.4) | 62.39 ± 26.69 | | 8.11 ± 4.31 | |
| Age | Less than 25 | 33 (29.2) | 66.73 ± 27.31 | 0.272 (0.846) | 7.21 ± 4.95 | 0.313 (0.821) |
| | From 25 to less than 35 | 39 (34.5) | 65.54 ± 18.13 | | 7.03 ± 3.54 | |
| | From 35 to less than 45 | 18 (15.9) | 63.44 ± 26.27 | | 7.22 ± 3.93 | |
| | More than 45 | 23 (20.4) | 61.04 ± 29.03 | | 8.09 ± 4.80 | |
| Education | Secondary education | 15 (13.3) | 55.87 ± 33.44 | 1.119 (0.330) | 8.80 ± 4.97 | 1.054 (0.352) |
| | University education | 61 (54.0) | 66.20 ± 23.69 | | 7.18 ± 4.33 | |
| | Postgraduate studies | 37 (32.7) | 65.62 ± 21.39 | | 6.97 ± 3.87 | |
| Household residents | From 1 to 3 | 38 (33.6) | 64.11 ± 24.67 | 0.340 (0.797) | 7.47 ± 4.22 | 0.589 (0.624) |
| | From 4 to 6 | 56 (49.6) | 66.18 ± 25.38 | | 7.09 ± 4.57 | |
| | From 7 to 9 | 14 (12.4) | 59.00 ± 25.36 | | 8.43 ± 3.80 | |
| | 10 or more | 5 (4.4) | 67.20 ± 7.01 | | 5.80 ± 2.17 | |
| Abiding by social distancing | Low | 17 (15.0) | 60.59 ± 22.79 | 0.300 (0.742) | 8.35 ± 4.53 | 0.674 (0.512) |
| | Moderate | 34 (30.1) | 66.18 ± 21.23 | | 7.41 ± 3.77 | |
| | High | 62 (54.9) | 64.90 ± 26.73 | | 7.00 ± 4.48 | |
| COVID-19 infection of friends family and/or coworkers | Yes | 21 (18.6) | 65.67 ± 24.22 | 0.942 (0.348) [b] | 7.16 ± 4.12 | 0.855 (0.395) [b] |
| | No | 92 (81.4) | 60.10 ± 25.70 | | 8.05 ± 4.95 | |

SD standard deviation; NSS Nursing Stress Scale; BRCS Brief Resilient Coping Scale. [a] One-way ANOVA test. [b] Independent samples *t* test.

Considering the Nursing Stress Scale and Brief Resilience Coping Scale, Table 2 (Figure 2) shows a severe stress level in 50.4% of the studied sample compared to 41.6% who reported average coping abilities. Additionally, there was a highly statistically strong negative linear significant correlation between the NSS and BRCS ($p \leq 0.01$).

As shown in Table 3, a moderate stress level was reported among studied nurses in all Nursing Stress Scale subscales (death and dying 61.9%, conflicts with physicians 53.1%, inadequate preparation 66.4%, lack of support 50.4%, conflicts with other nurses 60.2%, and workload 39.8%); except for the subscale uncertainty concerning treatment, mild stress was reported (54.0%). A highly statistically significant correlation was found between the Nursing Stress Scale subscales and the scale total score and the Brief Resilience Coping Scale's total score ($p < 0.01$). Except for the subscale uncertainty concerning treatment, the subscale had statistically significant total scores of the Nursing Stress Scale ($p = 0.040$) and non-statistically significant total scores of the Brief Resilient Coping Scale ($p = 0.431$).

**Table 2.** Total Score of Nursing Stress Scale and Brief Resilience Coping Scale among Studied Nurses. (*n* = 113).

| | | No. (Percentage) | $r_s$ (Sig.) |
|---|---|---|---|
| Nursing Stress Scale (NSS) | Mild | 13 (11.5) | |
| | Moderate | 43 (38.1) | |
| | Sever | 57 (50.4) | −0.837 (0.000 **) |
| Brief Resilient Coping Scale (BRCS) | Poor | 42 (37.2) | |
| | Average | 47 (41.6) | |
| | Good | 24 (21.2) | |

$r_s$: Spearman's Correlation Coefficient test. ** Highly statistically significant at $p \leq 0.01$.

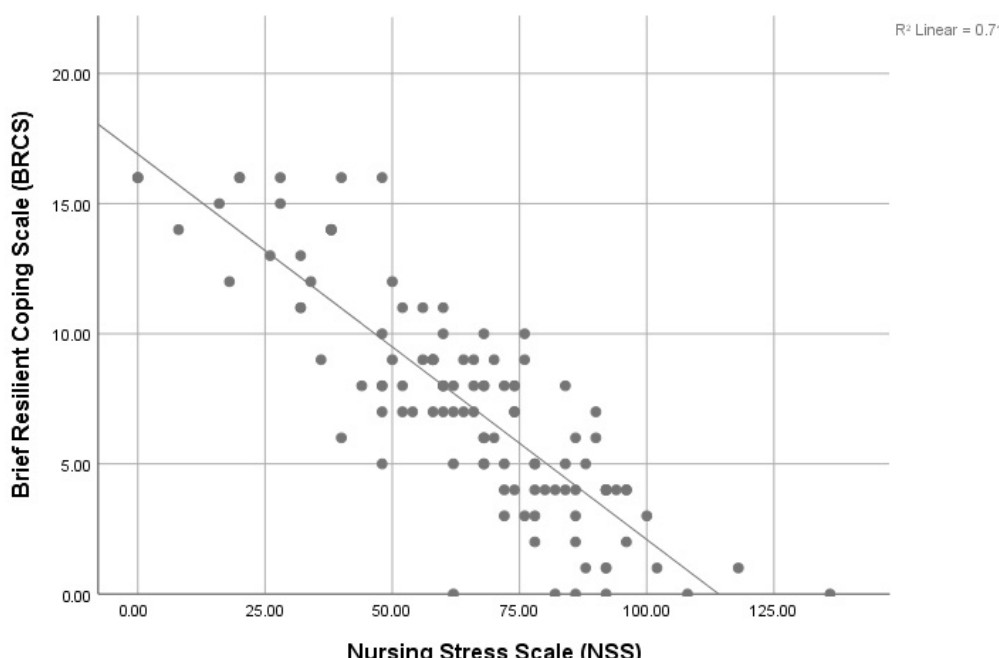

**Figure 2.** Scatterplot between Nursing Stress Scale and Brief Resilient Coping Scale among Studied Subjects.

**Table 3.** Nursing Stress Scale Subscales Score and Correlation with Total Score and Brief Resilient Coping Scale.

| NSS Subscale | No. (Percentage) | | | Test (Sig.) [a] | |
|---|---|---|---|---|---|
| | Mild | Moderate | Sever | NSS | BRCS |
| Death and dying | 21 (18.6) | 70 (61.9) | 22 (19.5) | 0.922 (0.000 **) | −0.914 (0.000 **) |
| Conflicts with physicians | 24 (21.2) | 60 (53.1) | 29 (25.7) | 0.923 (0.000 **) | −0.832 (0.000 **) |
| Inadequate preparation | 23 (20.4) | 75 (66.4) | 15 (13.3) | 0.887 (0.000 **) | −0.632 (0.000 **) |
| Lack of support | 24 (21.2) | 57 (50.4) | 32 (28.3) | 0.783 (0.000 **) | −0.675 (0.000 **) |
| Conflicts with other nurses | 36 (31.9) | 68 (60.2) | 9 (8.0) | 0.631 (0.000 **) | −0.307 (0.001 **) |
| Workload | 26 (23.0) | 45 (39.8) | 42 (37.2) | 0.857 (0.000 **) | −0.869 (0.000 **) |
| Uncertainty concerning treatment | 61 (54.0) | 32 (28.3) | 20 (17.7) | 0.193 (0.040 *) | 0.075 (0.431) |

[a] Spearman's Correlation Coefficient test. * Statistically significant at $p \leq 0.05$. ** Highly statistically significant at $p \leq 0.01$.

*3.2. Qualitative Data*

Regarding quantitative data, Table 4 shows coping strategies to moderate work-related stress as reported by studied nurses during qualitative data collection in the interview; 57.5% of them gave more attention to their spiritual side, 55.8% used communication with family members or friends, and 47.8% choose to accept the current situation. Additionally, 36.3% of the studied sample practiced a hobby, and the coping strategies least reported were seeking psychological support, reducing exposure to COVID-19-related information, and finding distractions (8.0%, 21.2%, and 28.3%, respectively).

**Table 4.** Coping Strategies as Reported by Studied Nurses to Deal with Work-related Stress. (*n* = 113).

| Coping Strategy Reported | No. (Percentage) * |
|---|---|
| Attention to the spiritual side | 65 (57.5) |
| Communicate with friends and family | 63 (55.8) |
| Accept the current situation | 54 (47.8) |
| Helping others in need | 41 (36.3) |
| Find things to do/distraction | 32 (28.3) |
| Reduce exposure to disease-related information | 24 (21.2) |
| Seek psychological support | 9 (8.0) |

* Numbers are not mutually exclusive.

Moreover, data showed that studied nurses at the beginning of the pandemic lacked the appropriate education and experience to deal with such emerging conditions. They had only a little relevant knowledge that could be useful in their practice, plus a minimal amount of information about what was happening in China, earlier in 2019, and later in half of the world. Nurse 054 said, "I knew that whatever happening in China will eventually reach us, and we are not prepared, and not even in a hundred years, we are not ready mentally or financially to face quarantine or social distancing." Just like most affected countries at first, they had neither the knowledge for a proper diagnosis nor an appropriate treatment protocol for affected cases. As the current study was conducted after the start of the pandemic, nurses' experiences were divided into three different themes: initial pandemic stress, family-related stress, and coping measures.

### 3.2.1. Initial Pandemic Stress

COVID-19 is now in Egypt, with an increasing number of cases and public panic. By the end of the year 2020, most governmental and university hospitals had been converted into quarantine areas to adapt to the increased number of cases, even though almost none of the hospitals were fully prepared to face this conversion. Nurses along with other medical staff were forced to work under inhumane conditions, facing death every day. Nurse 016 noted, "So, you receive one gown, one surgical mask, and one sterile glove per day, with eye protection or face shield that is released just one time, that you have to clean and disinfect yourself and reuse again till it wears out." The fact the personal protective equipment was scarce was frightening, and wearing PPE through a twelve-hour shift, with no chance of replacing any part of it, puts you at risk of contracting an infection every time you use the bathroom or have a break, eat, or drink anything during your shift. Nurse 087 added, "not even an N95 mask, only regular surgical mask, even while working in an ICU." Although not every patient requires high PPE to be used, at the beginning of the pandemic, no one was certain of anything, and the information on how much PPE was needed for each patient was not sufficient.

The reluctant use of PPE among medical personnel created uncertainty about if it is even important to use the full PPE preparation for each patient, as the World Health Organization (WHO) revealed that the mask is not effective to use, considering that SARS-CoV-2 in a heavy virus that cannot travel in the air, except for aerosol transmission [20]. Nurse 015 said, "I didn't wear a mask while working with patients in the outpatient clinics, they were mostly asymptomatic, and were never physically close to me, and I never had symptoms." No doubt that the increased number of deaths and infections within the healthcare team was a huge trigger of stress among other medical professionals, along with consequences and unknown complications of the new disease.

### 3.2.2. Family-Related Stress

A huge part of the stress generated was the fear of transmitting the virus outside the hospital to friends, family, or the community during commute, considering that some quarantine areas had twelve- to twenty-hour shifts, rather than the fourteen-day protocol. "With the first hospital admission of a suspected COVID-19 case, I could not go home, I spent the night at the nurses' locker room. I was afraid of my family, especially my

two daughters" Nurse 019 said. Nurse 098 added, "my grandparents and my father have diabetes, I could survive a COVID infection, but none of them would." The anxiety associated with viral transmission to others was common among all nurses, forcing them to take extreme measures to ensure they would not pass the infection to others. Nurse 062 said, "I leave my shoes at the doorstep, change my cloth in the bathroom, wash them separately daily, and a hot water bath is mandatory before I start to interact with any family member."

Again, the community had their part in stigmatizing healthcare workers during the pandemic: "people in our street were covering their faces when I pass them by" Nurse 65 said. Some families applied an isolation protocol for family members working in hospitals, and other nurses utilized self-quarantine or self-isolation to save their families. Nurse 087 stated, "During the early months, I used to sleep in a separate room by the house door and eat with disposable plastic kitchenware." Nurse 013 added, "my family was afraid of me, when I get home, they don't spend any time or get near me."

### 3.2.3. Coping Measures

The spiritual dimension was the dominant determinant of coping among studied nurses. Considering the religious nature of the Egyptian community, nurses reported mostly relying on God and religion in passing hard times, among other stuff. Nurse 004 said, "I delegate my affair to God and seek His help and do my job taking the necessary precautions and then relying on God." Nurse 081 noted, "Getting closer to God and taking reasons, and nothing will happen to us except what has been decreed for us and written by God for us". Nurse 012 added, "...and coping with the current situation as a permanent situation." "Pray to God and grovel to him that this situation ends quickly," Nurse 070 said.

Others decided to accept the facts and face whatever comes, with Nurse 004 saying, "Whoever did not die by poisoning died of other than it, the causes are many, and death is one." Nurse 030 said, "focus on what I can change and let go of what I can't change." That came from the professional commitment to helping others, regardless of the self-risk. In addition, sometimes nurses were forced to return to work as soon as they recover from a COVID-19 infection, due to the nursing shortage and ethical responsibility. Nurse 101 noted, "I used to tell my friends, we have to work, it's an international crisis, if we didn't act now, it will get each one of us, even at home."

On the other hand, the studied nurses reported attention to family ties and reaching out to other family members as a means of coping with stress, to " . . . spend time with family . . . " as noted by Nurse 040 and, as mentioned by Nurse 064, to "Communicate with brothers, sisters, and relatives by phone." Nurse 016 remarked on connecting with community members, emphasizing the importance of helping others in need: " . . . Providing food commodities to those who are financially unable . . . ". Moreover, Nurse 080 said, "Family bonding helped reduce stress." As important as it is, having a high-speed internet connection was a factor in reducing stress during long working hours as a means to contact family and friends, get the latest information related to COVID-19, and so on. Nurse 094 said, "that was a great starting time for a lot of people". Through the internet, nurses passed the time connecting with their loved ones, learning about COVID-19, and discovering something new.

## 4. Discussion

The current study gathered information from 113 nurses working with COVID-19 patients. The study aimed to evaluate stress and explore coping strategies experienced by nurses during the COVID-19 pandemic. As the COVID-19 pandemic escalated into a global crisis in a short period of time, the current study results show no significant relationship between sociodemographic characteristics and the Nursing Stress Scale or coping strategies, which is considered normal as no previous experiences have occurred in the lifetime of most nurses in Egypt. The results are supported by a previous cross-sectional study on stress, anxiety, and depression among healthcare workers facing the COVID-19 pandemic

in Egypt [21], in which researchers found no relation between sociodemographic characteristics or stress, anxiety, or depression among participants. Another study [22], investigating the factors affecting stress among nurses providing care for COVID-19 isolation hospitals in Egypt, also revealed a nonsignificant relation between age, gender, and experience with stress level among participants.

Considering nursing stress, multiple studies showed a high level of stress among nurses [23–26] along with other medical staff members [27,28] during the COVID-19 pandemic. The current study revealed a severe-to-moderate stress level among the studied nurses, which is quite fair, considering the current fluctuation in new viral strains and diagnosed cases and the continuous shortage of healthcare workers as reported by the World Bank [29]. Along with previous studies, the current study results are indistinguishable from those of a previous study in Egypt [30] investigating the occupational stress and job satisfaction among nurses during the COVID-19 pandemic, revealing a high level of stress among participants, regardless of their working place (triage, isolation, and non-isolation hospitals).

Moreover, the studied nurses had a moderate stress level considering death and dying, conflicts with physicians, inadequate preparation, lack of support, conflicts with other nurses, and workload. In a previous study supporting the current study results [22], participants showed a moderate level of stress regarding the nursing stress subscale score. Additionally, another study on the COVID-19 pandemic impact on health professionals regarding stress, resilience, and depression [31] revealed registered nurses as the most stressed healthcare members.

Regarding resilient coping, the studied nurses reported an average-to-poor level of resilient coping during the pandemic; however, their coping was negatively correlated to their stress level. This could be related to the novelty of the COVID-19 situation. In agreement with the current study results, a study on the positive aspect of the COVID-19 trauma [32] and factors associated with post-traumatic growth [33] revealed that exposure to trauma promotes resilience and coping among nurses, including through the development of a positive attitude towards the problem, emotional and problem-focused attitudes, and self-reliance and self-care. Additionally, the study on stress, resilience, and coping of healthcare workers during the COVID-19 pandemic [34] revealed normal levels of coping, and higher levels of stress among nurses.

Furthermore, the studied nurses had an indisputable fear of infection transmission to their family, friends, or community, forcing them to apply extreme protective measures. Infecting others who might be a child, elderly, or chronic disease patient would counter the caring nature of most nurses. As mentioned in the previous study on nursing students' experiences [35], nursing students shared the same fear of spreading the infection to their families. Additionally, in a similar study in Iran [36], participants were concerned about their children, the elderly, and chronic illnesses in their family. The spiritual dimension was also dominant in coping with the emerging pandemic situation and stress, not only in the nurses in the current study, but also in similar studies [36–38], emphasizing the importance of the spiritual dimension during the pandemic, and the fact that many such social values have roots in religion, particularly those that creatively integrate religious and human notions and are brought to light in times of social unrest. The spiritual growth of nurses is influenced by their dedication to their religion, their devotion to ethics, and their commitment to the law.

Regarding other coping measures, the studied nurses revealed a variety of coping measures that are considered effective in the related literature [36,38], while others used various measures to moderate their stress, such as early experiences, food, and even alcohol and other recreational drugs [39]. Research correlated coping with higher resilience [40]. As a human response to stress, coping is an inherent quality that prevents emotional damage to the individuals.

Finally, a better understanding of the perceptions, attitudes, and experiences of nurses during the COVID-19 pandemic is provided by the study's findings. The information

gathered is useful in identifying the stressful demands and reoccurring stressful issues among nurses, and potential coping measures that could be useful to adapt in the future.

*Limitations of the Study*

The study also had some shortcomings. Because of the studied nurses' time limits and working conditions, some of them were reluctant to take part. Additionally, the generalization of study findings should be limited; the study's findings should not be applied to other cultures or professions, as they reflect the opinions and experiences of a small number of respondents. The extremely distinct professional identities of nurses and the elements of human-centered nursing theories that have influenced most of them to pick their profession should unquestionably be taken into consideration. Moreover, biases related to qualitative research such as information, selection, and confounding bias, etc., had little effect considering the mixed research design used. The researcher declares a potential bias though, taking into consideration their subjective viewpoint, their experiences, and judgment.

### 5. Conclusions

In conclusion, nurses who work with COVID-19 patients experienced elevated levels of stress regardless of their characteristics and eventually developed an adaptive response to cope with their stress. Spiritual coping strategies, communication with family, and helping others were the most dominant resilient coping approaches employed by nurses to moderate stress. Additionally, the higher the level of resilient coping, the less stress experienced by nurses. On a positive note, future similar experiences would be easier to handle and correlate to less stress if coping strategies and adaptive measures were readily available to employ at an early stage.

**Funding:** This research received no external funding.

**Institutional Review Board Statement:** Ethical review and approval were waived for this study due to the nature of the study, as it was an observational study.

**Informed Consent Statement:** Informed consent was obtained from all nurses involved in the study.

**Data Availability Statement:** Data is contained within the article.

**Acknowledgments:** The researcher thanks all nurses and other healthcare members everywhere in isolation hospitals and other healthcare facilities, who have served and are still serving as frontline defenders protecting patients and the community from COVID-19 and other illnesses.

**Conflicts of Interest:** The author declares no conflict of interest.

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
