# Peer review of "Stress and Resilient Coping among Nurses: Lessons Learned from the COVID-19 Pandemic"

_psych, doi:10.3390/psych4030047_

Round 1

Reviewer 1 Report

The current study showed the stress condition of nurses, as well as how their Resilient Coping during COVID19. They well-discussed their outcomes and limitations. Their conclusions suggested that good resilient coping approaches contributed to the stress management.

One comment on their study design is that they measure stress condition of nurses during COVID19 pandemic. However, none of their results compare the stress level before and after the pandemic. Or nurses who work with COVID19 patients or not. They should not conclude that nurses experience elevated levels of stress based on their results. They may discuss it based on the references.

Some minor comment as follows.

Page 2 line 70, why the authors said their study sample was “purposive”. Sample should be random picked. Although the author selected 113 nurses working in the specific time and location, they probably not mentioned “purposive”.

Page 7 line 224, “nurse 65” the number should be three digitals to keep consistent.

Author Response

Response to Reviewer 1 Comments

One comment on their study design is that they measure stress condition of nurses during COVID19 pandemic. However, none of their results compare the stress level before and after the pandemic. Or nurses who work with COVID19 patients or not. They should not conclude that nurses experience elevated levels of stress based on their results. They may discuss it based on the references.

Response 1: This could be addressed in a future systematic review or meta-analysis, comparing stress levels among nurses before and after the pandemic. Sadly we couldn't ask nurses to provide an old experience with stress compared to a highly stressful time like the current pandemic.

Page 2 line 70, why the authors said their study sample was “purposive”. Sample should be random picked. Although the author selected 113 nurses working in the specific time and location, they probably not mentioned “purposive”.

Response 2: The availability of a vast population for a random pick is not available unfortunately for the author. So, the sample is either a purposive or convenience sample, we believe a purposive sampling would be more convenient than a convenience sampling technique, considering the research aim. 

Page 7 line 224, “nurse 65” the number should be three digitals to keep consistent.

Response 3: Corrected

Reviewer 2 Report

This study and its results aligns well with similar studies for other parts of the world. It is important for current scientific literature to include studies and responses from different sociocultural and economic backgrounds, for the global scientific community to understand how stressful situations affects the healthcare providers in different settings.

On a minor note, it would have been prudent to include or describe some of the inherent bias due to the study design, such as non-response bias or researcher bias in the study limitations.

Although out of the purview for the current study design, a small description on how to mitigate the effects of stressful situations future directions for similar situations could add on to the conclusion statement.

Author Response

Response to Reviewer 2 Comments

On a minor note, it would have been prudent to include or describe some of the inherent bias due to the study design, such as non-response bias or researcher bias in the study limitations.

Response 1: Added the following statement to the study limitations: “Moreover, biases related to qualitative research such as information, selection and confounding bias … etc. had little effect considering the mixed research design used. The researcher declares a potential bias though, taking into consideration their subjective viewpoint, their experiences, and judgment.”

Although out of the purview for the current study design, a small description on how to mitigate the effects of stressful situations future directions for similar situations could add on to the conclusion statement.

Response 2: Added the following statement to the study conclusion: On a positive note, future similar experiences would be easier to handle and correlate to less stress if coping strategies and adaptive measures were readily available to employ at an early stage.

Reviewer 3 Report

Overall an interesting study assessing the stressors in healthcare during a pandemic. It is unclear what this paper adds to the current body of knowledge. It would be beneficial in your discussion and conclusions to demonstrate why this study is important and how it will contribute to the health and safety of nurses moving forward in healthcare. You have briefly touched on this but it could be better clarified and described. The paper is generally clear - some language editing is needed. Some inconsistencies have been noted. The results are well presented. The discussion is fair but more emphasis on the "why" would be valuable. Also, how it may translate to other regions. There is no ethics statement included. 

Be consistent with wording. You use "coronavirus", "covid19" and "COVID-19" throughout. 

Line 49" Should be "use" not uses. 

Line 58: Your method should be written in past tense. 

Line 58: Your aim should be part of the introduction, not the methods. 

Line 60: Figure 1 is missing. 

Line 71: You do not need to state "previously mentioned workplace and timeframe". 

Line 74: How does a lack of time and setting influence how they completed the survey? Are you saying that it was delivered online due to ease of access? 

Line 75: What social media platforms were used and how? Did the differences in methods for interviews create any differences in your results? 

Line 77: Why did a physical illness exclude people from the study? Who determined this? 

Line 79: What is the interview questionnaire? Is this the online questionnaire? You mention qualitative questions but this is a quantitative data collection tool? It is very unclear what you are referring to in this section. It would be beneficial to separate these section into sub-headings "Questionnaire" and "Interview", clearly distinguishing between the quantitative and qualitative components.

Line 106: Why did you use an old version of SPSS? Version 28 is the latest version. 

Line 112: If the questionnaires had already been validated, why were you using Cronbach's alpha? 

Line 125: What do you mean by "non-statistically" significant? 

Line 132: Figure 1 should be in the methods section where it is mentioned. 

Line 138: Figure 2 is not mentioned in text. 

Line 165: You are discussing quantitative data here but the heading is qualitative data. 

Author Response

Response to Reviewer 3 Comments

Overall an interesting study assessing the stressors in healthcare during a pandemic. It is unclear what this paper adds to the current body of knowledge. It would be beneficial in your discussion and conclusions to demonstrate why this study is important and how it will contribute to the health and safety of nurses moving forward in healthcare. You have briefly touched on this but it could be better clarified and described. The paper is generally clear - some language editing is needed. Some inconsistencies have been noted. The results are well presented. The discussion is fair but more emphasis on the "why" would be valuable. Also, how it may translate to other regions. There is no ethics statement included.

Response 1: Corrected.

Be consistent with wording. You use "coronavirus", "covid19" and "COVID-19" throughout.

Response 2: Corrected throughout the article “Covid19”.

Line 49" Should be "use" not uses.

Response 3: Corrected.

Line 58: Your method should be written in past tense.

Response 4: Corrected.

Line 58: Your aim should be part of the introduction, not the methods.

Response 5: In the authors' template download from the site, the site mentioned that aim should be included at the end of the introduction section. The aim in the methods section was included in similar studies published in the Psych journal.

Line 60: Figure 1 is missing.

Response 6: Figure 1 is not missing, it is the descriptive figure of the convergent triangulation model (available in line 134 page 4).

Line 71: You do not need to state "previously mentioned workplace and timeframe".

Response 7: Corrected.

Line 74: How does a lack of time and setting influence how they completed the survey? Are you saying that it was delivered online due to ease of access?

Response 8: it was delivered online for the following reasons, 1) lack of time for nurses during and after shifts, 2) lack of a meeting place at the hospital, 3) ease of access; distribution, collection, and tabulation of results, and 4) Eco-friendly.

Line 75: What social media platforms were used and how? Did the differences in methods for interviews create any differences in your results?

Response 9: Whatsapp and Facebook voice calls mainly; phone calls were also considered for some participants. The variation of communication means was not an issue or a potential bias, because it’s all almost similar in the way of delivering information between the researcher and participants.

Line 77: Why did a physical illness exclude people from the study? Who determined this?

Response 10: Physical or mental, acute or chronic illness could generate stress on its own, even greater stress with the ongoing pandemic. Such potential stress could generate some bias in results, so it was excluded from the study to prevent potential bias.

Line 79: What is the interview questionnaire? Is this the online questionnaire? You mention qualitative questions but this is a quantitative data collection tool? It is very unclear what you are referring to in this section. It would be beneficial to separate these section into sub-headings "Questionnaire" and "Interview", clearly distinguishing between the quantitative and qualitative components.

Response 11: Corrected
Subsection 2.3. Data collection tools, has 4 bullets for 3 tools, and the 4th is questions guide to quantitative data collection.
Sub-subsection 2.3.1. quantitative data collection tools

Sub-subsection 2.3.2. qualitative data collection tools

Line 106: Why did you use an old version of SPSS? Version 28 is the latest version.

Response 12: (it was SPSS version 25) According to the IBM release on the SPSS version 28 (https://www.ibm.com/downloads/cas/DKA95AXM) the changes are almost with minimal significance to nursing research; all required statistical tests for the current study were not changed from the 25th to the 28th version. Moreover, the best version to use according to statistical blogs is version 19.

Line 112: If the questionnaires had already been validated, why were you using Cronbach's alpha?

Response 13: Chronbach’s Alpha was reported to vary widely among studies and across countries to the sample standardised tool.

Line 125: What do you mean by "non-statistically" significant?

Response 14: p-value >0.05, there was no change in stress or coping between different sociodemographic characteristics subgroups (i.e. males/females).

Line 132: Figure 1 should be in the methods section where it is mentioned.

Response 15: Corrected, moved under its first-time mention in the methods section (line 63 page 2).

Line 138: Figure 2 is not mentioned in text.

Response 16: Corrected (mentioned in line 141 page 4).

Line 165: You are discussing quantitative data here but the heading is qualitative data.

Response 17: Table 4 was generated from the qualitative data, should it be considered quantitative?

Round 2

Reviewer 3 Report

Thank you for the revised manuscript. You have addressed most comments very well. The aim should still be moved to the introduction (your response also indicates this) and Line 125 should say "not statistically significant" rather than non-statistically significant as per the response to this query. The latter term means something different.

There are still some major issues with your qual vs quant data collection definitions.

Line 110: This subheading says quantitative but you are talking about qualitative. Line 177  and 178 are still very confused. Your sub-heading is qualitative data but the paragraph is about quantitative data. Is Table 4 qualitative (subjective, experiential data - e.g. from an interview) or is it quantitative (survey questions)? This will need to be amended. 

Well done on a great study. 

Author Response

Response to Reviewer 3 Comments

ROUND 2

Thank you for the revised manuscript. You have addressed most comments very well. The aim should still be moved to the introduction (your response also indicates this)

Response 1: Aim is already mentioned in the last paragraph as requested in the PSYCH template, I added the word aim so it is more clear. The aim in the methods was an imitation of a similar study I found published in the same journal, but I removed it from the methods section anyways.

Line 125 should say "not statistically significant" rather than non-statistically significant as per the response to this query. The latter term means something different.

Response 2: Corrected (in line 122) – (line 119 in PDF).

There are still some major issues with your qual vs quant data collection definitions.

Line 110: This subheading says quantitative but you are talking about qualitative.

Response 3: Corrected (2.3.2. Qualitative data collection tools).

Line 177 and 178 are still very confused. Your sub-heading is qualitative data but the paragraph is about quantitative data. Is Table 4 qualitative (subjective, experiential data - e.g. from an interview) or is it quantitative (survey questions)? This will need to be amended.

Response 4: Table 4 was generated from the qualitative data collected during the interview and from open-ended survey questions, therefore placed in the qualitative section.

Added the sentence (lines 180 and 181) – (lines 173 and 174 in PDF) “Regarding quantitative data, table 4 shows coping strategies to moderate work-related stress as reported by studied nurses during qualitative data collection in the interview, …” to minimize confusion.

Or … I can move it to the quantitative section -upon your request- if that would make more sense of the data.
